# Methodological Issues in Analyzing Real-World Longitudinal Occupational Health Data: A Useful Guide to Approaching the Topic

**DOI:** 10.3390/ijerph19127023

**Published:** 2022-06-08

**Authors:** Rémi Colin-Chevalier, Frédéric Dutheil, Sébastien Cambier, Samuel Dewavrin, Thomas Cornet, Julien Steven Baker, Bruno Pereira

**Affiliations:** 1CNRS, LaPSCo, Physiological and Psychosocial Stress, Université Clermont Auvergne, F-63000 Clermont-Ferrand, France; frederic.dutheil@uca.fr; 2Preventive and Occupational Medicine, CHU Clermont-Ferrand, F-63000 Clermont-Ferrand, France; 3Wittyfit, F-75000 Paris, France; samuel.dewavrin@wittyfit.com (S.D.); thomas.cornet@wittyfit.com (T.C.); 4Biostatistics Unit, The Clinical Research and Innovation Direction, University Hospital of Clermont-Ferrand, CHU Clermont-Ferrand, F-63000 Clermont-Ferrand, France; scambier@chu-clermontferrand.fr (S.C.); bpereira@chu-clermontferrand.fr (B.P.); 5Centre for Health and Exercise Science Research, Hong Kong Baptist University, Kowloon Tong, Hong Kong 999077, China; jsbaker@hkbu.edu.hk

**Keywords:** methodological issues, modeling, occupational health, longitudinal data, missing data, cluster-correlated data, real-world data

## Abstract

Ever greater technological advances and democratization of digital tools such as computers and smartphones offer researchers new possibilities to collect large amounts of health data in order to conduct clinical research. Such data, called real-world data, appears to be a perfect complement to traditional randomized clinical trials and has become more important in health decisions. Due to its longitudinal nature, real-world data is subject to specific and well-known methodological issues, namely issues with the analysis of cluster-correlated data, missing data and longitudinal data itself. These concepts have been widely discussed in the literature and many methods and solutions have been proposed to cope with these issues. As examples, mixed and trajectory models have been developed to explore longitudinal data sets, imputation methods can resolve missing data issues, and multilevel models facilitate the treatment of cluster-correlated data. Nevertheless, the analysis of real-world longitudinal occupational health data remains difficult, especially when the methodological challenges overlap. The purpose of this article is to present various solutions developed in the literature to deal with cluster-correlated data, missing data and longitudinal data, sometimes overlapped, in an occupational health context. The novelty and usefulness of our approach is supported by a step-by-step search strategy and an example from the Wittyfit database, which is an epidemiological database of occupational health data. Therefore, we hope that this article will facilitate the work of researchers in the field and improve the accuracy of future studies.

## 1. Introduction

In previous decades, randomized clinical trials have been the main experimental methodology used to collect clinical data. Collected within a strict framework, the analysis of randomized clinical trials data is used to answer specific questions related to a specific population (namely the population selected for the study). This remains true provided that analyses are performed rigorously with consideration for all required hypotheses and methods (e.g., definition of a representative population sample, randomization, controlled tests, application of appropriate models to the data). However, the advances in technology, the democratization of computer tools such as computers and smartphones as well as the increase in data storage capacities offer researchers new possibilities for collecting large amounts of health-related data for analysis. Such data, collected independently of traditional trials, is also known as real-world data [1,2,3]. More broadly, the data can be obtained from various sources such as patient health records, product and disease registers or even digital health data collection platforms and applications. Many examples of real-world databases can be listed and classified according to the sources from which they come. As examples of this, we can cite records from patient registries such as the European Cystic Fibrosis Society Registry database [4], healthcare databases such as Wittyfit and Wittyfit Research [5], pharmacy and health insurance databases such as the Food and Drug Administration’s Sentinel Initiative [6], social media such as PatientsLikeMe [7] or patient-powered research networks such as PCORnet [8]. Big data management approaches such as the management of chronic kidney disease [9], chronic obstructive pulmonary disease [10], or lymphoma subtypes [11] have naturally emerged from such databases. 

As a perfect complement to randomized clinical trials data and presenting a longitudinal structure [12], the results of the analysis of real-world data, also called real-world evidence, are very popular; for example, within pharmacology research, these data sets are considered essential because they are crucial in pharmacological decision-making [13,14]. However, by construction, real-world databases are not immune to missing, noisy (outliers), duplicate or inconsistent data phenomena [15].

A longitudinal study is a study investigating repeated data, i.e., where the same data has been measured/collected on an individual several times over time [16]. Unlike cross-sectional studies, cohort effects and time-related effects can be measured separately. The main goal of a longitudinal study is to describe changes over time and measure the individual influence of variables to explain changes observed [17]. Longitudinal data may also facilitate the change over time when investigating particular individuals [18]. For example, it is possible to measure risk factors in the development of disease for particular individuals in the population [19]. However, many challenges and methodological issues make it difficult to analyze longitudinal data. These analytical problems include the correlated structure of intra-individual data, the considerable size of data sets, irregular time-spaced measurements, non-linear patterns (such as rapid growth or stationary responses), latent constructs, and the mix of time-varying and static covariates. 

There are also more difficult problems to consider, such as cluster-correlated data, missing data and longitudinal modeling itself [20,21]. To sum up, studying such data types requires understanding the specific concepts that are outlined in Figure 1. 

Although a multitude of approaches have been proposed in the literature, it is still difficult to account for all these concerns in a synchronized way. The purpose of this paper is to review the state of the art of different methods and models designed to address issues related to real-world longitudinal occupational health data in a holistic way to improve the quality of future surveys on the topic. In the first part, we aim to introduce each methodological issue and to present advanced methods for addressing them. In the second part, we aim to present a case study in which all the issues overlap and present a method to meet the objectives of the study using the solutions previously featured.

## 2. Methodological Issues and Methods for the Analysis of Longitudinal Data

### 2.1. Cluster-Correlated Data

Cluster-correlated data occur when pre-existing groups of individuals exist within the population, allowing for a natural classification of subjects into groups, otherwise known as clusters. Thus, the observations of the same cluster are correlated with each other while the observations between the different clusters are uncorrelated. This structure requires considering two possible sources of variability, namely intra- and inter-cluster variability. These similarities in the data can, however, lead to a loss of statistical efficiency and integrity of the models used, necessitating an increase in the size of the sample [22]. It is therefore necessary to find an appropriate balance between the number of clusters and the number of patients.

Correlated data occur in a number of different settings. They can be clustered, spatial (grouping by location), multilevel or longitudinal data or even several at the same time [23]. A hierarchical or multilevel structure can be thought of as a series of several levels nested within each other. Thus, the lowest level represents level 1, and the levels follow each other upwards to the level where all the data is contained. The units formed at each level can then be likened to clusters. A hierarchical structure is therefore a succession of more and more specific clusters and, from observation of the lower levels, is a structure that is “neither accidental nor ignorable” [24]. Multilevel structures widen the field of possibilities in terms of questions that can be answered (e.g., presence of a group effect, disparities within the different clusters). On the other hand, the structures question the use of statistical methods that do not consider the nature of the structures, which could also provide erroneous results [25,26]. Longitudinal data is a special form of clustered data. By nature, an individual’s data is plausibly more positively correlated with each other than with other individuals. Like correlated data, this intra-subject correlation must be taken into account in the analyses, otherwise this can provide false positive results and erroneous confidence intervals [21,27]. Figure 2 presents an example of a population divided into clusters followed in a longitudinal study. 

Thus, cluster-correlated data, regardless of its type, requires special analytical treatment [28]. Cluster-correlated data analysis attempts to take into account the variability associated with each level of the structure and must be interpreted separately from the overall variability [29]. Indeed, misspecification of the cluster effect or careless interpretation of the model parameters can lead to erroneous results [30,31]. Hence, in the literature, multilevel models (MLMs) have been designed to address these various issues and analyze correlated-clustered data [24,29,32,33]. These models are similar to mixed models. In fact, the “levels” described by the multilevel models can be assimilated to be the “random effects” of the mixed model. This implies that cluster-correlated data can be analyzed properly using the models defined above by considering the levels of the data as random effects. Similarly, for longitudinal data, this implies that the individual effect must be considered as a random effect also.

Clustered data are not immune to the phenomenon of missing data. In the absence of data, conventional imputation methods can be applied. However, the particular structure of cluster-correlated data, which leads to multiple sources of variability (intra- and inter-cluster), suggests proceeding differently. As a result, it appears preferable to apply the imputation method chosen cluster by cluster [34]. To sum up, in the presence of cluster-correlated data, it is preferable to estimate the missing values within clusters. When the data of an individual is missing, estimates can be made using individuals from the same cluster. For a hierarchical structure, the lowest level cluster in which the individual is located will be preferred. If an entire cluster leaves the study, the missing data can be estimated from the data of individuals at the level above. Finally, at the highest level of the hierarchy, it can be estimated using the entire data set. 

### 2.2. Missing Data

The phenomenon of missing data is a common problem in data analysis, especially in longitudinal surveys. It is rare for a data set to be complete, particularly for long studies, either because of occasional omissions or because of subject withdrawal. As a consequence, data missingness causes three main issues [35]: it can introduce a significant amount of bias, make it more difficult to process and analyze the data and induce a loss of statistical power [36,37]. When faced with missing data, the most common used methods include the complete case analysis (or listwise deletion) and the “last observation carried forward” methods. However, these methods are too hazardous and can introduce a significant bias in the estimation of the parameter under investigation [38,39,40]. As previously observed, these ad hoc methods are no longer desirable or necessary. Likewise, in the presence of clustered data, it is difficult to estimate what effect the loss of an individual or even an entire cluster has on the outcome measures. Accordingly, it is better not to ignore missing values but to try to estimate them using imputation methods. The purpose of these methods is to estimate missing values from previously observed values, which can be considered as measured data and therefore used for fitting analytical models. While these methods may seem attractive, their use is not without problems [41]. Even though there are some methods that are more effective than others, it is not really possible to say which method is especially good [42]. These methods require understanding and apprehension of certain concepts that we aim to outline below.

To be more effective and valid, statistical analysis must be performed with suitable mechanisms and assumptions for missing data [37]. In other words, it is crucial to understand the process behind it to ensure the validity of the statistical inferences and the absence of bias [43]. Therefore, a careful preparation should be observed before using a method that is very similar to an identification process. In particular, it is essential to identify the two main parameters: the structure and the type of missing data. 

The structure plays an important role in the choice of imputation methods that can be applied [44]. There are three possible structures, which can overlap. The structure is said to be univariate if one and only one variable of an individual has missing values. It is said to be monotonic when several variables of an individual have missing values and these variables can be classified according to their percentage. For longitudinal surveys, this structure can be found when an individual leaves the study, for example in case of attrition or even abandonment. Attrition, or random drop-out [45], is a major and common problem in longitudinal studies, representing an important challenge when modeling [46] as it may produce estimates of the effects that are often underestimated [47]. If it is not possible to obtain the data of an individual at a given time (e.g., forgotten appointment, forgotten connection), we often consider intermittent monotonic structures. The structures can qualify as arbitrary (or non-monotonic) if they are neither univariate nor monotonic. Figure 3 illustrates the different structures of missing data.

After identifying the structure of missing data, it is necessary to deal with the identification of its type. We consider here again three different types: data which is missing completely at random (MCAR), missing at random (MAR) and missing not at random (MNAR) [41]. A missing value is considered to be MCAR if its missingness does not depend on other observations, whether they are observed or not. Under this assumption, the results of the analyses appear to be generally unbiased, but it is rarely verified because it is restrictive. A less restrictive assumption supposes that if it depends on observed observations only, its type will be MAR. Otherwise, if it is neither MCAR nor MAR, it will be MNAR. Despite this formulation, it is not always possible in practice to identify the causes of data missingness and therefore to determine its type [48]. 

Finally, once the type and structure of the missing data are identified, an imputation method can be used. Table 1 provides a non-exhaustive overview of the different methods that can be used to calculate estimates of missing values [41,49]. 

For the particular case of MNAR type data, it is necessary to perform a ”sensitivity analysis”. Indeed, although there is a multitude of models to analyze this type of data, they all rely on assumptions (e.g., the missing data mechanism) which, for this type of missing data, are unverifiable. When the results are particularly sensitive to the assumptions made, it becomes difficult to choose the most suitable model. That is, instead of fitting a single model, it is wiser to consider alternative models and to assess the sensitivity of the results to the assumptions made about the missing data mechanism [31,41,48].

### 2.3. Longitudinal Data and Modeling

Repeated data models have been the subject of many years of research and development. Today, the literature on the subject is substantial and many of these models have been used in longitudinal surveys. Among the models used most for the analysis for longitudinal data [20,50], the most often used are variance models, whether univariate (ANOVA) or multivariate (MANOVA), random effects models such as mixed models or generalized linear models (GLMs) using generalized estimating equations. Nevertheless, other models such as trajectory models or alternatives to linear and mixed models such as structural equation models (SEMs) or “cross-lagged” panel models (CLPMs) have appeared in the literature and constitute promising alternatives for the analysis of longitudinal data. These models are not all similar and are intended to answer different questions. While mixed models and their alternatives are used to estimate the impact of factors on a response variable, trajectory models are used to model individual trajectories within a population and follow their evolution over time. Given a large amount of data, modeling should be performed on a training sample and model validation on a test sample. We will now describe the above-mentioned models in more detail. 

#### 2.3.1. Analysis of Variance for Repeated Measures

The analysis of variance for repeated measures represents a classical method to analyze longitudinal data. Whether for ANOVA or MANOVA, they allow comparing groups’ means on a dependent variable across time. However, it does not allow learning about individual trajectories. In addition to other parameters, time is also treated as an explicative variable. This means, among other things, that these models assume that individuals cannot have a proper slope over time, which is rarely true. Moreover, these models are difficult to apply in the absence of data and are applied using the complete cases or the “last observation carried forward” methods. As a consequence, it is therefore best to avoid these models and to turn to more suitable methods for repeated data analysis [27]. For example, mixed models offer more advantages [36], especially by adding a subject-specific component to the model and allowing the conduct of the analysis despite possible lack of data.

#### 2.3.2. Mixed Models

Mixed models can be seen as natural extensions of regression models [51,52]. Unlike the latter, they involve random effects specific to each individual. The mean and the variance of the response are respectively modeled as a linear combination of fixed-effect and random-effect components, where the impact of the factors is weighted and associated with the coefficients of the model [53,54]. Unlike analysis of variance models, both marginal and mixed models are unconditional. This feature makes it possible to model the response variable as a function of both the covariates and time separately representing both within- and between-subject effects [20]. They are also particularly effective for making individual predictions even despite eventual missing data because the randomly missing estimates are unbiased [41]. More broadly, generalized linear mixed models combine the specificities of generalized linear models (GLMs) and mixed models, allowing for generalizing the type of the explained variable. This change in construction is ideal for the analysis of longitudinal data structures since it makes it possible to account for the intra-subject correlation through random effects [55,56].

This makes mixed models a safe and effective method for the analysis of longitudinal data. Other more specific models, as we will see below, have been designed subsequently and added to the literature, but mixed models remain the most popular. Once a trajectory is identified, it is possible to describe it over time but also in terms of static covariates using a mixed model or equivalent.

#### 2.3.3. Generalized Estimating Equations

In general, the estimation of GLM parameters is based on the maximum likelihood method [57]. Yet, the generalized estimating equations method does represent a popular alternative to likelihood-based generalized linear mixed models, allowing the model to be extended to the analysis of correlated data. Afterwards, they simultaneously model the link between explicative variables, the response and the within-subject dependence. This method is particularly appreciated on the one hand because it gives efficient and unbiased estimates of the parameters of a generalized linear model and on the other hand because it accounts for intra-individual correlation [58,59]. Generalized estimating equations can deal with missing data under the assumptions that they are MCAR. When these data are MAR, the estimates might be biased [60].

#### 2.3.4. SEMs and CLPMs, Complementary Approaches to the Mixed Model

Despite its effectiveness, the mixed model is only a first step towards more complex statistical methods allowing the move from a global analysis of the population to a more personalized/individualized analysis of occupational data. Therefore, models such as SEMs or CLPMs can be used as alternatives to mixed models for the analysis of longitudinal data. 

SEMs can be viewed as a much more comprehensive regression method, including dependent and independent variables. Where SEMs stand out is in the ability to additionally take into account hypothetical latent constructs, and to examine relationships between observed variables and these concepts [61,62,63,64,65]. As such, they can be seen as a natural combination between factor analysis and regression or path analysis. Although they are particularly powerful, SEMs stay very sensitive to the problem of missing values. In addition to the assumptions of normality and independence that they require, SEMs need a very large sample size to fit. On average, at least 500 individuals and up to >2500 individuals if one of the assumptions is not verified are needed to expect good estimates [65]. 

CLPMs are used to describe and estimate reciprocal relationships or directional influences between longitudinal variables [66,67]. Cross-analysis is particularly used to describe causal relationships between variables. Often contested because of their low statistical power and their limitations (e.g., the need for a large sample of longitudinal data, the stationarity and synchronicity assumptions or the causal relationship assertion), simpler methods such as multiple regression are often preferred [68]. Because of this, alternative models to SEMs such as the random intercept “cross-lagged” panel models have been developed to overcome their shortcomings [69]. In addition to the temporal stability assumption of the classical method, these consider the trait-type stability invariant of individuals over time. 

#### 2.3.5. Trajectory Models

By plotting longitudinal data, we obtain curves otherwise called individual trajectories or developmental trajectories. A developmental trajectory describes and provides information on the evolution of an individual over time. A trajectory is used to describe a latent process that cannot be observed directly but can be explained over time using measured (therefore observed) variables from which its trajectory is inferred [70,71]. These methods that take into account unobserved heterogeneity are called person-centered [72]. Nowadays, several common methods [73,74] are used: generalizations of mixed effects models such as growth curve models [75], growth mixture models [76,77], group-based trajectory models [78,79,80,81], latent class analysis [82,83] and latent transition analysis [72].

Most of these methods identify several different trajectories within the population, also called latent classes. For each trajectory, the estimated share of individuals in the population belonging to it is given, which reflects its shape, and each individual has a certain probability of belonging to each trajectory. Individuals are then assigned to the group corresponding to the trajectory for which the probability of belonging is the highest. Once a trajectory is identified, it is possible to explain its shape over time but also in terms of static covariates using a mixed model or equivalent.

Although an individual trajectory can be estimated despite prospective missing values, it is difficult to conclude on the validity of this trajectory, especially when data is NMAR [84]. It is essential to have a sufficient amount of data for an individual, otherwise it is impossible to determine its trajectory [85]. These models are also considerably sensitive to assumptions, and misspecification can lead to biased estimates for trajectories and an overestimation of the number of classes by the model [73]. 

## 3. Case Report

### 3.1. Introduction

We aim here to illustrate our suggestions with a complete and illustrated example using the Wittyfit database [5]. Here, we focus on a small part of the approaches described above, namely the contribution of trajectory models to mixed models in a real-world longitudinal occupational health data framework. 

Let us imagine that we want to analyze the annual evolution in job satisfaction of workers at different companies between 2018 and 2021, and to observe if the gender and the job position of a worker can affect job satisfaction. A first look at the objective allows us to confirm that we are in the presence of clustered (multiple companies), correlated (individual effect) and longitudinal (follow-up over time) data. If a worker did not express his or her sentiment in a year, he or she is assigned a missing value. Thus, we are faced with the main methodological issues mentioned in this paper.

### 3.2. Methods

#### 3.2.1. Participants and Exlusion Criteria

Wittyfit software is a web-based platform designed to assess workers’ health through a holistic approach of the individual. Volunteers are invited to express their feelings on different health-related outcomes using visual analog scales. With more than 40,000 active users in about nearly 80 companies, whose first registrations began in January 2018, the Wittyfit database offers researchers a substantial behavioral and longitudinal database to study the evolution of workers’ occupational health through various indicators such as job satisfaction and stress. Workers not present at the baseline (2018) or with too few data [85] were excluded. 

#### 3.2.2. Outcomes

Job satisfaction of workers was assessed using a related visual analog scale, scaled from 0 to 100. Workers could rate their personal feeling of job satisfaction as many times as they wanted. A worker’s overall annual job satisfaction score was computed as the average of the notes that he or she filled in over the year.

Socio-demographic characteristics of workers (gender and job position) were filled in by corporates clients of Wittyfit. The job position of a worker is defined according to whether he is an employee or a manager. 

#### 3.2.3. Statistics

Statistical analyses were performed using R (version 4.1.1) in the RStudio (version 1.4.1717) platform. The imputation of the data was done using “longitudinalData” (with the ‘locf’ and ‘linear.interpol’ methods) and “mice” packages. Group-based trajectory modeling was realized with the “latrend” package. We combine the commands ‘lcMethodLcmmGBTM’ and ‘lcMethods’ to define the model and the ‘latrendBatch’ command to fit it, with nonstructured matrix of variance-covariance. Model assumptions (residual independence and normality, and variance homogeneity) were verified a posteriori. Unless specified, we considered a *p*-value < 0.05 as statistically significant for analyses. 

### 3.3. Data Application

First, we need to analyze the data structure to determine if it has a multilevel structure in addition to a longitudinal structure itself. In our example, we count multiple companies to which workers belong. Therefore, this observation indicates to consider the companies as clusters. We now need to focus on missing data by analyzing its type and structure. The missing data presents two different patterns: (1) a non-monotonic (e.g., due to a worker who did not express in a year) and (2) a non-monotonic pattern (e.g., due to a worker who did stop expressing over the years). For the second one, as data is longitudinal, a drop-out phenomenon may occur. Since it is reasonable to assume that an individual’s drop-out may depend on unobserved data (e.g., for an individual who quit during the studied period because of a lack of job satisfaction), we need to suppose that the drop-out is informative, and therefore that the type of missing values is MNAR. This assumption should therefore lead us to perform a sensitivity analysis by applying several imputation methods, for example with the “last observation carried forward” and the linear regression interpolation methods, but also using multiple imputation, taking into account the company effect. 

To study the evolution of workers’ job satisfaction, we can first apply a linear mixed model assessing the influence of time on job satisfaction (Figure 4a). Starting with a single trajectory given by the mixed model, we can observe whether different evolutions exist in the population. To do this, we can apply a trajectory model such as the GBTM on the original dataset and confirm the results using the imputed datasets. The model allows us to identify five latent classes within the population, with a minimal average posterior probability of assignment of 70.5% and minimal odds of correct classification of 5.5, thus exceeding the classical thresholds of 70% and 5 expected for this type of model [79] (Figure 4b). 

Finally, to explore the relationship between a worker’s sociodemographic characteristics and job satisfaction, we can apply a generalized mixed model with the trajectory to which the worker belongs as the outcome and the company effect as a random effect. Thus, according to the model, both gender (β = 1.60, 95% CI 1.15 to 2.24, *p* = 0.005) and job position affect the job satisfaction of the worker (β = 5.49, 95% CI 2.72 to 13.13, *p* < 0.001). This result remains true regardless of the dataset, non-imputed or imputed. 

### 3.4. Conclusions

From a self-imposed context, we presented here the many challenges we had to face in analyzing real-world longitudinal occupational health data. This study had two objectives, namely the identification of the existence of different evolutionary trajectories of job satisfaction within a population of workers and the role of the job position on the level of job satisfaction. Each methodological issue has been raised and addressed using the appropriate methods in agreement with the purpose of our article, allowing us to meet the aims of the study.

## 4. Conclusions

As a direct result of the rise of massive databases [5,6,7,8] and advances in computational and digital tools and their democratization, real-world data analysis has allowed researchers to confirm the results of previous randomized clinical trials and provide new knowledge, known as real-world evidence [1,3]. The progressive and incessant accumulation of data thus offers the possibility of collecting health-related, longitudinal, individual data, all at a low cost, thus allowing the development of data-analysis-centered medical approaches.

Although the analysis of longitudinal data has become widespread over the last 30 years, particularly through mixed models, and has made it possible to study population evolution, this article demonstrates that methodological issues remain and have begun to find an echo in more recent approaches, allowing for a move towards a more predictive, preventive, personalized and participatory medicine. Despite the advances offered by these new approaches, some of these issues remain to be addressed before proceeding with the analyses [20,21], in addition to the usual issues linked to any data analysis, limiting their use and overall understanding [74].

In this article, we aimed to present the state of the art of the methods developed for the analysis of real-world longitudinal occupational health data while exposing the three main issues encountered during the analyses of such type of data, namely the concepts of cluster-correlated data, missing data and longitudinal data itself. In addition, we have provided an example of data analysis presenting these issues and discussed the steps to be taken in the analysis process. A search strategy using numerous general article databases was conducted, showing that there is currently no approach to deal with these three methodological issues. 

We believe this article can serve as a practical guide for future studies on the topic to improve experimental quality, especially for non-statistician researchers who wish to keep abreast of the new approaches available and the issues involved. Future studies will focus on the comparison between the different models and approaches presented here. Simulation work has already been done [86,87,88,89], but not in this specific framework.

## Figures and Tables

**Figure 1 ijerph-19-07023-f001:**
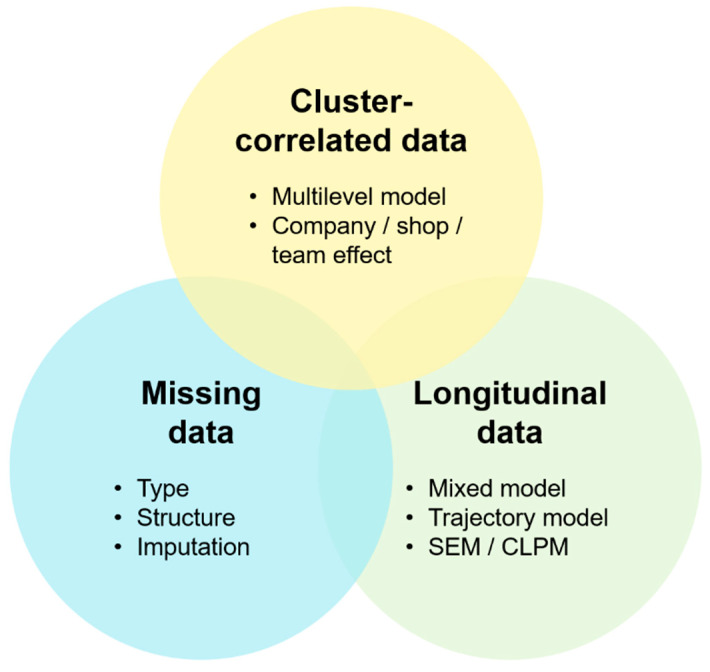
Data diversity of real-world longitudinal occupational health databases (see Appendix A for the novelty and usefulness of our approach and main formulas surrounding cluster-correlated data, missing data, and longitudinal data). SEM: structural equation model, CLPM: “cross-lagged” panel model.

**Figure 2 ijerph-19-07023-f002:**
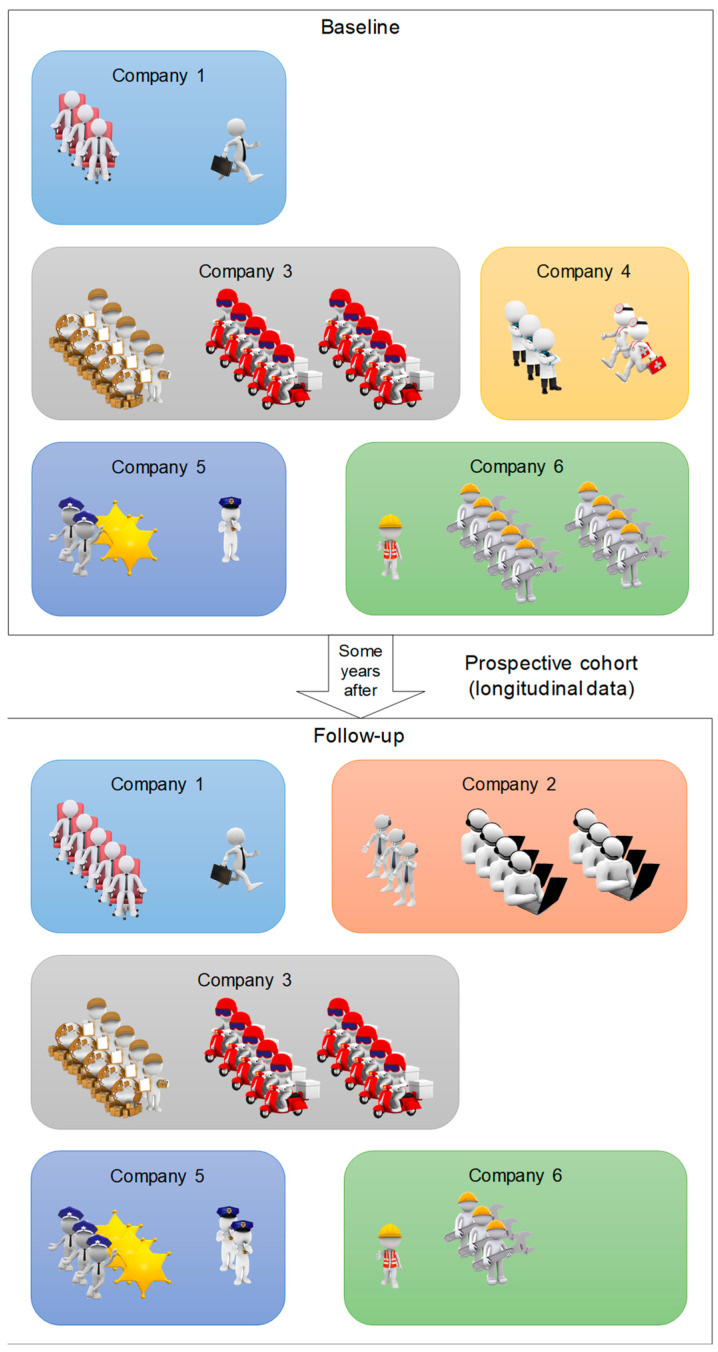
Example of cluster-correlated data: here, a prospective cohort of workers from multiple companies is followed over time.

**Figure 3 ijerph-19-07023-f003:**
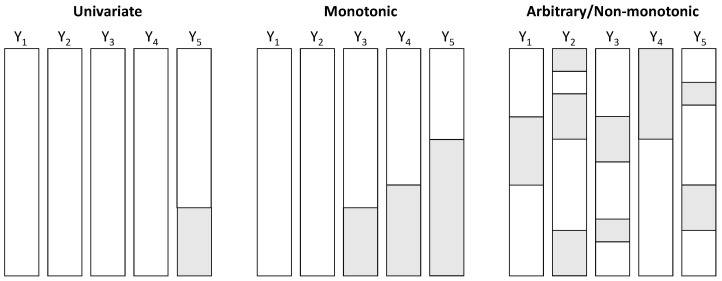
Missing data structures. White sections indicate the presence of data, shaded sections their absence.

**Figure 4 ijerph-19-07023-f004:**
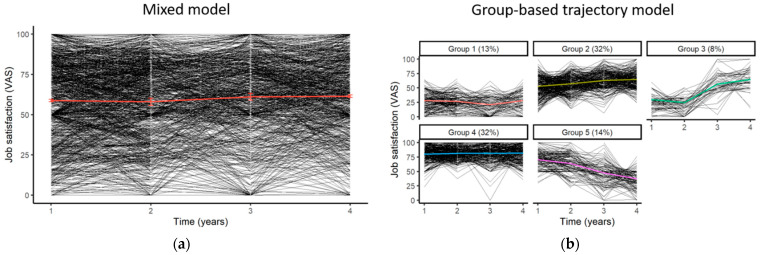
Evolutionary trajectories of job satisfaction among Wittyfit users: (**a**) mean trajectory using a mixed model (standard errors of coefficient are symbolized by error bars), (**b**) individual trajectories using a group-based trajectory model, where each group represent a possible evolution of the workers’ job satisfaction in the population (e.g., individuals belonging to “Group 3” are characterized by a slight decrease between times 1 and 2, then by a sharp increase beyond time 2).

**Table 1 ijerph-19-07023-t001:** Imputation methods by type of missing data.

Missing Completely at Random		Missing at Random	Missing Not at Random
**Ad hoc methods** Complete case analysis, available-case analysis, weighting methods		Expectation maximization algorithm	“Sensitivity analysis”
**Single imputation**			
**Implicit modeling** Hot/cold deck imputation, substitution, composite methods	**Explicit modeling** Mean/regression/stochastic regression imputation	Multiple imputation	

## Data Availability

Data from Wittyfit cannot be transmitted without the prior consent of the company’s corporate clients, except to the University Hospital of Clermont-Ferrand, France, which may use the data for research purposes.

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
