# Peer review of "Methodological Issues in Analyzing Real-World Longitudinal Occupational Health Data: A Useful Guide to Approaching the Topic"

_ijerph, 2022, doi:10.3390/ijerph19127023_

Round 1
Reviewer 1 Report
1. Novelty of the paper must be explained more in comparison the similar research.
2. Model validation is completed.
Reviewer 2 Report
Very good paper
We suggest only enlarging the literature review highlighting similar big data management approaches in other domains such as the approach proposed in ENHANCING TRAVELLER EXPERIENCE IN INTEGRATED MOBILITY SERVICES VIA BIG SOCIAL DATA ANALYTICS - ScienceDirect, Technological Forecasting and Social Change
Please clarify the motivation of the study and write a conclusion section of exemplificatory nature
Reviewer 3 Report
In the first and in the second review, I have very clearly stated the following necessary conditions to accept the paper:
1. A very detailed, formal, methodological description of the methods allowing reproducibility of the analysis.
2. Reproducible data application – Readers must be informed what specific models are used, what are the assumptions and if they are met, what verification procedures were used etc.
3. The Monte Carlo simulation study described in the first review.
Re 1. In this version of the paper, Authors added very, very general descriptions of many, many methods which can be used in the considered analysis, but the description is not directly connected with the data application, and it does not allow to reproduce the analysis.
Re 2. The data application is completely not reproducible. Still, I do not know what specific models are used, what are the assumptions and if they are met, what verification procedures were used.
Re 3. No work on Monte Carlo analysis, described in the first review, has been done.
The Authors do not meet the necessary requirements stated in the last two reviews. In my opinion the paper should be rejected.
Reviewer 4 Report
Reviewer comments to "Methodological issues to analyze real-world longitudinal occupational health data: a useful guide to approaching the topic" by Colin-Chevalier et al.
This is an improved version of a previously reviewed manuscript. I have to say that authors significantly improved the previous version, attending most of my previous suggestions.
Moreover, the article is now submitted in the modality of research note, which makes it more adequate for the proposed structure. The appendix sections reviews some information that were lacking in previous revisions.
The descriptions of the figures and the figures themselves were also improved.
I just have some minor comments before a possible acceptance of the paper in IJERPH:
C1. Figure 1. Figures should appear as soon as they are mentioned (inside the corresponding topic).
C2. Figure 3 shows some interest results; however, the explanation of the data trends observed is insufficient. I suggest to explain better in the text these results (i.e., meaning of errorbars and A-F subplots). Perhaps, it may be convenient to name the figures as (a) and (b) (this is optional).
C3. 5. Patents section is empty. Please check if this section is necessary or not.
C4. It is not common to repeat a figure and a legend in an article. Here, Figure 1 is repeated. Perhaps, it should be convenient to only reference the former figure that appears at the beginning of the manuscript.
C5. Appendix A. Please check if subtopics and equations in the appendix should be numerated.
C6. Figure A2. It is not clear in the figure what is the correspondence between the squares and circles. Perhaps, adding a line to relate the information could help.
Author Response
Please see the attachment.

This manuscript is a resubmission of an earlier submission. The following is a list of the peer review reports and author responses from that submission.
Round 1
Reviewer 1 Report
In my opinion, the paper is publishable after some minor corrections as follows:
- English language and style are fine/minor spell check required.
- The novelty of the paper must be highlighted in Abstract.
Reviewer 2 Report
The paper is very good and deserves to be accepted after minor revisions are made.
The main issue is with the absence of a proper description of the case report, with mathematical and formal details. Please upgrade this section and resubmit.
Reviewer 3 Report
Major issues
The paper will not be interesting for readers familiar with the methods, because it does not cover any new methodological issues. It will be also not of interest to practitioners who would like to use these methods in their research – it does not explain even standard methods in detail.
I expect that papers published in journals with high IF present new, topical ideas and proposals - this paper does not contain new proposals. The Authors state (l. 27-29), that “Although solutions exist to meet these data collection challenges, solutions are not always correctly exploited, especially in cases where data collection issues overlap. In an attempt to solve this problem, we have conceived a process that considers all three issues simultaneously”. In Google Scholar I have found more than 13 000 papers where the following phrases are used “longitudinal” and “imputation” and “mixed model” which for sure covers simultaneously all the issues of Authors’ interest not exhausting the topic at the same time. What is more, I think we can assume that in most of them the topic is “correctly exploited”.
If the Authors do not present their own, new proposals (as in this paper) I expect at least the following elements in the paper.
- A very detailed, formal, methodological description should be presented, including a formal description of the details of the methods declared to be discussed (including formal criteria of the choice of the methods, formal description/definitions of the terms used in the paper, formal description of algorithms and their statistical properties, etc.).
- A reproducible application with a clear description should be included, the choice of the methods should be justified by formal procedures, and valuable conclusions covering both practical and theoretical issues should be presented.
- Due to the complexity of the defined problem, the paper should include an extensive, reproducible Monte Carlo simulation study covering different NA scenarios and different model scenarios where the problem of the misspecification of the chosen method should also be included showing e.g., a possible decrease of the goodness-of-fit or estimation/prediction accuracy due to the incorrect specification of the model or NA treatment.
At least, to accept the paper with no original scientific/research/methodological proposals all of these three components should be included, but none of these issues has been covered by the Authors. The paper should be rejected.
Additional remarks
- Longitudinal data can be population data, a probability sample, and a non-probability sample. This implies different methods which should be used to analyze such data, different sources of possible randomness of estimates or predictions, different methods to be used to assess estimation or prediction accuracy. These fundamental issues have been ignored by the Authors.
- Authors claim that “This article reviews the various solutions proposed and attempts to analyze all three in detail.” (l. 26) while no methodological details have been presented.
- l. 33-34: “(…) and provide results with higher confidence levels.” — a researcher can always increase the confidence level – it is not the aim. The aim is to increase accuracy (under the assumed confidence level).
- l. 42-43 – how do you define a “representative conclusion”?
- l. 43-44 – the sentence “This remains true provided that the analyses are performed rigorously with consideration for all required hypotheses.” is unclear from the methodological point of view.
- The information value of Table 1 is very, very limited – it does not explain any issues for readers who are not familiar with these topics. Many terms used in Table 1 have not been explained and discussed, none of them has been discussed exhaustively. The same problem occurs in the case of Figure 1. Similarly, Figure 2, which is almost one page large and – in my opinion – does not include any valuable, scientific, or methodological details including its description.
- l. 204-105: “It can be classified in four different types: clustered, spatial (grouping by location), multilevel or longitudinal data” – the classification is unclear. These types are not “different”, as stated by the Authors – the longitudinal data can be multilevel and spatial, and clustered.
- Although Authors identify some very standard issues which can occur in the considered analyses, their proposed solutions are so general that they are not useful in practice, see e.g., l.165-167: “Therefore, careful preparation should be observed before using a method that is very similar to an identification process. In particular, it is essential to identify the two main parameters: the structure of the data and also the type of NA”.
- l. 309-310 no information about possible different random effects, criteria of the choice of the model, its goodness-of-fit, results of the tests of significance of fixed effects and variance components, no information about the properties of these tests and their assumptions (are they met for the dataset?).
- l. 313-314 – the decision of the type of NAs is based on the Authors’ believes (“which leads us to believe that the structure of NA will be mainly monotonic”), not formal methods which stand in contradiction to the Authors’ statement presented in lines 27-29: “Although solutions exist to meet these data collection challenges, solutions are not always correctly exploited, especially in cases where data collection issues overlap. In an attempt to solve this problem, we have conceived a process that considers all three issues simultaneously.”
- l. 319-334 – the description of the last part of the analysis is unclear and devoid of methodological details, the Authors present a conclusion with no practical or theoretical implications on the description of the phenomenon: “The model allows us to identify four trajectories. (…) a “rising” trajectory (…) a “downward” trajectory (…) a “high” trajectory (…) a “low” trajectory” (…)” The same conclusion can be made for many other datasets!
- The lack of the description of the considered database.
Editorial remarks
- The bibliography is not prepared carefully, see e.g., l. 421 and l. 422.
- l. 232 – The sentence should start with a capital letter.
- l. 276 – “modelS”
- l. 361, 366, 369 – the text should not be indented.
Reviewer 4 Report
Reviewer comments to "Methodological issues to analyse real-world longitudinal occupational health data" by Colin Chevalier, R., et al. This is a review paper that explains the main methodological issues that can occur when analyzing real-world longitudinal occupational health data. The main problems are defined as: the longitudinal data itself, not available (missing) data, and cluster-correlated data. Moreover, the authors provided a test case to demonstrate the effect of different models for repeated data using a modified data set. I consider that this is an interest paper that could provide some guidance to analyze data from different sources. Although the review is focused on health data, some concepts could be applicable to different areas of research. In my opinion, the topic is interesting and the writing style is acceptable; however, the structure requires some updates and it is necessary to include some information to make the results reproducible. As a REVIEW paper, it should follow the MDPI PRISMA guidelines for structure and presentation of the research. I do not recommend the publication in IJERPH in the current form. My comments are listed below.
MAJOR COMMENTS
C1. Lines 75-77. "studying such data types requires understanding the specific concepts that are outlined in Table 1". This table requires more explanation in the main text.
C2. Table 1. It is not clear for the reader what MCAR, MAR and NMAR mean. Perhaps, a brief definition in the main text or in the table could help. It is important to mention that all abbreviations should be explained in the main text as soon as they appear.
C3. Line 81. "the issues outlined". This is a new paragraph, then, it may be unclear to the reader to identify the referred issues.If you are referring to previous paragraph, thus, to join both phrases may help.
C4. I suggest to explain the current increase in knowledge that your work is presenting with respect to previous research. Even when this is a review paper, a more detailed description of the contribution of your work is required.
C5. Line 84. The topic 2 "Methodological issues and methods....." starts just after the purpose of the paper, in the Introduction, without previous guidance to the reader. I recommend providing a brief structure of the work. Please note that a Review paper should follow the MDPI guidelines to define the structure of the paper. For example, these guidelines suggest to specify the METHODS employed to obtain the data, the number and type of documents that were investigated, the procedure to perform the research, etc. Please organize your work to accomplish with the MDPI PRISMA guidelines for REVIEW articles. Link: https://www.mdpi.com/editorial_process#standards
C6. Line 85, 143. Why the subtopics in Section 2 do not follow the sequence that was described in the Abstract?: longitudinal data itself, missing data, and cluster-correlated data. I suggest presenting the information systematically along the paper, since this is expected by the reader, unless the reader is guided to a new content/structure (for this reason is recommended to follow the PRISMA guidelines).
C7. Line 319. "we can apply a trajectory model ". More details about the method to obtain Figure 3 are needed to allow the reader to reproduce your results. Results should be reproducible. Please provide enough details to do so.
C8. Lines 323-325. Is this phrase the conclusion about your analysis? It is not clear what is the objective of the analysis and if this finding will be useful to the reader. Please improve the discussion of the results.
C9. Line 342. Conclusions. Please provide conclusions regarding the case study that was evaluated and the main problems of analysis that were reviewed.
MINOR COMMENTS
C10. Figure 1. This figure needs more explanation in the main text.
C11. Line 100. Figure 2 needs to be better introduced/described in the main text.
C12. Line 127. "describes". Please check.
C13. Line 169. "Three different structures need to be considered." Which ones? Please clarify.
C14. Line 194. "unverifiable assumptions". what kind of assumptions? Please explain.
C15. Lines 149 & 199. "2.2" is repeated. Does the subtopic "Modeling" have the same relevance as the previous subtopics (Cluster-correlated data and Missing data)?. Please revise the organization of subtopics.
C16. Lines 203-207. As you are presenting a review paper, thus, it would be convenient to provide some reference works for the specific models that were described.
C17. Lines 213-215. It seems that this phrase is out of place. Please check.
C18. Figure 3. I suggest to explain what is the meaning of the gray lines in the figure. It is not clear to the reader how you defined the Class1, 2, 3 and 4 trajectories. Please improve this.
C19. Appendix A. Please verify if the "Appendix A" should be mentioned in the main text. As far as I know, it has to complement the information described in the main text.
Round 2
Reviewer 3 Report
It is still the paper with no original scientific/research/methodological proposals. Most of the introduced changes are editorial ones.
I defined clearly necessary conditions to accept the paper in the previous review.
- A very detailed, formal, methodological description of the methods is still not presented. The Authors added the Appendix with the general description of the methods, but the description is not directly connected with the consecutive steps of the presented application (and it does not allow to reproduce the analysis).
- The data application is still not reproducible. In many places we are informed what can be done, but we do not know what has been done (e.g., “we can apply several suitable imputation methods”, “we will need to conduct a sensitivity analysis, for example by (…)”). Still we do not know what models are used, what are the assumptions and if they are met, what verification procedures were used etc.
- Although it was required, the Monte Carlo simulation study was not prepared.
What is more, the Authors claim that their paper is very novel because it jointly takes into account 4 topics, and (see l. 429-431 in the new version of the paper):
“Lastly, we found only n=1 article using the keywords “occupational” AND “cluster” AND “missing data” AND “longitudinal”. Therefore, there is currently no approach to dealing with the methodological issues to analyse real-world longitudinal occupational health data.”
I have checked this combination of words in google scholar and I have found not 1 paper (found by the Authors in PubMed) but more than 22 000 papers meeting Author’s criteria. Please check:
https://scholar.google.com/scholar?hl=eng&as_sdt=0%2C5&q=%E2%80%9Coccupational%E2%80%9D+AND+%E2%80%9Ccluster%E2%80%9D+AND+%E2%80%9Cmissing+data%E2%80%9D+AND+%E2%80%9Clongitudinal%22&btnG=
The Authors did not introduce the changes clearly defined by me in the previous review. This paper still neither explains the topic in details from theoretical point of view nor clearly presents the real data application allowing reproducibility. In my opinion, the paper should be rejected.
Reviewer 4 Report
Authors attended some issues of previous revision and improved the manuscript. However, I still have some concerns regarding the structure of the paper if it is considered as a Review work. I found repeated information and the structure of the work is not clearly defined. For this reviewer, the manuscript requires improvements and is not ready to be published in IJERPH as a Review article.
Some major comments are given as follows:
C1. Redaction of last paragraph of the introduction needs improvement. For example: "data types".... "this type of data"....."this type of data" again.. Furthermore, the gap in current research and the contribution of your work in this part of the paper should be included.
C2. The reader is not introduced to the structure of the paper from the Introduction or in a Method Section. In a previous revision, I suggested to consider MDPI PRISMA guidelines to improve this. I was expecting to find the organization of sections related to: Introduction, Methods, Results, Discussion, Conclusion, Appendix, or similar, which are recommended to report scientific research. PRISMA guidelines also propose a similar structure for review articles. As far as I know, authors aim to publish a review paper, then, these guidelines can help to better organize the ideas. I do not know if MDPI allows the current structure to be published as a Review paper. Perhaps, in case that authors want to keep the current organization of topics, an alternative could be to publish their manuscript as a Research Note or a Short Communication, which are also published by the journal (see https://www.mdpi.com/journal/ijerph/about).
C3. The subsections of Section 2 need to be checked. 2.2 is repeated. Please verify if the information corresponds to the proposed structure of subtopics. I commented this in the previous research; however, it was not corrected.
C4. Line 321. "the Wittyfit database". I suggest to provide a reference of the database just after it is mentioned for first time.
C5. Line 374. "the many challenges". This phrase should be verified. Lines 374-379 present a long phrase. It should be improved.
C6. Line 389. In the conclusions, it was mentioned that PubMed was used in the research; however, I could not find any method description referring to this in the main text. In my previous research I suggest to use the MDPI PRISMA guidelines to organize the work. It is common that all methods employed appear in the main text, perhaps in a main section that shows the Methods employed to 1) review the articles related to the objective of your research and 2) to explained the methods followed in the implementation of the study case. If it is intended to publish the research as a technical paper or a review article, I suggest to improve the organization of the work following the recommendations that were previously suggested.
C7. Line 408: "Supplementary materials". As 'supplementary materials' I was expecting complementary data provided in a database or similar. However, I found the same Figure 1 and information related to methods. If I am not wrong, this section seems like an Appendix instead of Supplementary Material. This should be verified.
C8. Lines 429-431. I wonder if the information in lines 420-432 is relevant to be included in a scientific paper. Is this information part of the Methodology? or is it the procedure to ensure that your research in new? It is not clear from the main text what is the reason of this information.
C8. Conclusions' description need to be reviewed.